# When Do LLMs Listen?: Confidence-Guided Knowledge Acceptance in LLMs

## Abstract

Large Language Models (LLMs) exhibit remarkable performance across a wide range of natural language tasks but face limitations in accessing dynamic or domain-specific knowledge not encountered during pre-training. To address this, recent research has explored integrating structured external knowledge from Knowledge Graphs (KGs) into LLMs via in-context learning (ICL). Although KG-augmented in-context reasoning has shown strong performance on commonsense tasks such as multiple-choice question answering (MCQA), the extent of LLMs' dependence on external knowledge remains poorly understood. Prior work has primarily examined which knowledge to extract from KGs and how to represent it to optimize prompts and task accuracy. In this study, we shift the focus to if and when LLMs accept or disregard injected knowledge. We introduce a confidence-guided framework that stratifies model predictions into high-, moderate-, and low-certainty bands: high means the model assigns dominant probability to a single choice, moderate reflects a few competing options with similar weight, and low corresponds to diffuse distributions with no clear preference. We then examine how knowledge interventions reshape probability distributions over candidate answers. Interventions include (i) supportive knowledge reinforcing the model's initial choice, (ii) opposing knowledge aligned with alternative answers, and (iii) noisy off-topic statements. Our analysis reveals systematic patterns: highly confident predictions tend to disregard opposing or noisy evidence, whereas moderate- and low-confidence predictions are more susceptible to change when exposed to contextual information. In particular, the model is most likely to switch between answer choices to which it has previously assigned similar mid-level confidence, while low-confidence options may gain probability mass but rarely enough to overturn the final decision. Noisy knowledge, however, induces only minor changes in the confidence distribution across choices. By moving beyond accuracy to behavioral response, this work provides a principled view of LLM robustness to knowledge augmentation and highlights design considerations for effective KG-enhanced question answering.

## 1 Introduction

Large Language Models (LLMs) have demonstrated remarkable capabilities across diverse natural language tasks, yet they face inherent limitations in accessing and utilizing specialized knowledge not explicitly encoded during training. These models are fundamentally bounded by their parametric knowledge, information embedded within their weights during pre-training, which may be incomplete, outdated, or insufficient for domain-specific reasoning tasks Brown et al. (2020). In-context learning (ICL) has emerged as a promising approach to address these knowledge limitations by embedding relevant information directly within prompts, allowing LLMs to leverage external knowledge without requiring model parameter updates Ren & Liu (2025); Liu et al. (2023); Dong et al. (2022). Despite the effectiveness of this approach, challenges remain in determining which information should (or not) be included in prompts to maximize performance.

Previous work has demonstrated that injecting external knowledge from knowledge graphs (KGs) can support the reasoning ability of LLMs in multiple choice question answering (MCQA) tasks. Many prior studies have focused on developing methods to incorporate such knowledge Yasunaga et al. (2021b); Luo et al. (2024a); Shen et al. (2020); Wang et al. (2020); Zhang et al. (2019); LUO

et al. (2024); Saxena et al. (2020); Markowitz et al. (2024); Sun et al. (2024); Zhao et al. (2025); Jiang et al. (2024). KGs provide structured representations of entities and their relationships, offering factual precision that unstructured text often lacks. By incorporating KG-derived information, LLMs can reduce incorrect reasoning and hallucinations without the need for costly model updates.

Nevertheless, in some cases, simply injecting knowledge from KGs that are related to the question-answering task into prompts is insufficient to improve performance. This limitation is particularly evident in commonsense reasoning tasks, where questions often require information from multiple KG subgraphs. In such cases, correct responses rarely follow from a single subgraph; instead, they depend on the combined evidence provided by several moderately relevant subgraphs that collectively support a coherent answer. Recent studies have shown that these models are sensitive to the availability, quality and presentation of external knowledge. While prior work has focused on identifying which knowledge to extract from KGs and how to represent it to improve performance, a question remains largely unexplored: when does a model actually accept or disregard the injected knowledge?

In this paper, we investigate model behavior through its preference over answer options. Specifically, we pursue three main goals. First, we study the confidence–stratified impact by partitioning questions according to the model's baseline confidence into low, moderate, and high categories, and quantifying how injected knowledge modifies the belief distribution over possible answers within each group. Second, we examine the effects of targeted knowledge by analyzing, for each confidence category, the influence of (i) supporting knowledge aligned with the model's initial prediction and (ii) competing knowledge aligned with alternative answers. This analysis captures both the probability shifts for individual choices and the frequency with which the top–1 prediction changes. Finally, we assess the model's robustness to noise by studying how off–topic or noisy knowledge affects its confidence in the original prediction, focusing on overall probability drift.

By shifting the focus from accuracy to behavioral response, our study aims to provide a systematic understanding of how large language models integrate, reject, or ignore external knowledge under varying levels of certainty. We argue that this perspective is crucial for diagnosing model limitations, guiding knowledge injection strategies, and ultimately informing the design of more robust and effective KG-augmented question answering systems.

## 2 RELATED WORK

LLM-based QA methods fall into two broad paradigms: those that rely solely on the model's internal knowledge Liu et al. (2022); Kojima et al. (2022); Wang et al. (2023), and those that incorporate external sources, notably KGs, to compensate for parametric limitations. Knowledge Graph Question Answering (KGQA) enhances factual accuracy and reasoning by combining textual and structured knowledge from KGs. Existing approaches to integrating KGs with LLMs can be grouped into four main categories:

**Knowledge Injection during Pre-training or Fine-tuning** These methods encode KG information into the model weights Shen et al. (2020); Wang et al. (2020); Zhang et al. (2019). Models like KGT5 Saxena et al. (2022) and RoG LUO et al. (2024) fine-tune on KG-grounded QA data, aligning outputs with KG structures.

**Embedding-based Methods** These represent KG entities and relations in latent space, using architectures tailored for reasoning. Early efforts like KV-Mem Miller et al. (2016) evolved into models like EmbedKGQA Saxena et al. (2020), QA-GNN Yasunaga et al. (2021b), GreaseLM Zhang et al. (2022), and TransferNet Shi et al. (2021). Though effective, these methods often require rigid, task-specific designs with limited flexibility.

**Semantic Parsing (SP)** SP-based approaches translate questions into logical forms (e.g., SPARQL) that are executed over the KG Sun et al. (2020); Lan & Jiang (2020). These enable precise symbolic reasoning but depend heavily on annotated data Das et al. (2021).

**Retrieval-Augmented Methods** These retrieve relevant KG subgraphs during inference and append them to the input prompt Yang et al. (2024); Li et al. (2023); Baek et al. (2023). PoG Tan et al. (2025) constructs multi-hop paths and prunes irrelevant facts; GCR Luo et al. (2024b) integrates KG structure via graph-constrained decoding. Some methods frame LLMs as agents that explore KGs iteratively Markowitz et al. (2024); Sun et al. (2024); Zhao et al. (2025); Jiang et al. (2024), though

this introduces latency Dehghan et al. (2024). While retrieval-based methods are model-agnostic and computationally efficient, their success depends on retrieving semantically relevant and complete facts. Poor retrieval can inject noise, harming answer quality.

# 3 PROBLEM DEFINITION

We formalize the multiple-choice question answering (MCQA) task as follows. Let each instance be represented as a tuple $(q, \mathcal{C})$, where $q$ denotes the question stem and $\mathcal{C} = \{c_1, \ldots, c_5\}$ is the set of five answer candidates. Given a text prompt $\Pi$ presenting $q$ with the five labeled choices, the model produces a normalized probability distribution $\mathbf{p} \in \Delta^4$ over the candidate set.[1]

We define the base distribution as $\mathbf{p}^{(0)} = (p_1^{(0)}, \ldots, p_5^{(0)})$, the base top-1 prediction as $\hat{y}^{(0)} = \arg\max_i p_i^{(0)}$, and the corresponding base confidence as $\kappa^{(0)} = \max_i p_i^{(0)}$. For each MCQA instance, the model's probability assignment induces a preference ordering from the most preferred (rank 1) to the least preferred (rank 5) candidate.

## 3.1 KNOWLEDGE GRAPH STRUCTURE AND EXTRACTION

A knowledge graph (KG) is defined as $G = (V, E)$, where $V$ is the set of entities and $E$ is the set of relations. Each edge $(h, r, t) \in E$, connecting a head entity $h$ to a tail entity $t$ via a relation $r$, represents a knowledge statement. The KG can thus be interpreted as a collection of structured statements encoding factual information about entities and their relationships.

Given a question $q$ and answer candidates $\mathcal{C}$, we ground entities from both the question and each candidate to the KG. Following prior work Yasunaga et al. (2021a), let $\mathcal{Q}(q)$ denote the set of grounded question concepts (QCs) and $\mathcal{A}(c_i)$ the answer concepts (ACs) for choice $c_i$. For each candidate $i \in \{1, \ldots, 5\}$, we extract all 1-hop and 2-hop paths connecting any QC in $\mathcal{Q}(q)$ to any AC in $\mathcal{A}(c_i)$:

$$\mathcal{P}_i = \{\text{paths of length } \leq 2 \text{ from } \mathcal{Q}(q) \text{ to } \mathcal{A}(c_i)\}.$$

These paths form our candidate knowledge statements. For noisy knowledge, we sample random 2-hop paths anywhere in the graph, unconstrained by question or answer concepts.

## 3.2 PROMPTING AND AGGREGATION

Each subgraph is linearized into textual form using edge labels and inverse labels when necessary. For example, a single edge might be rendered as *revolving_door (AtLocation) bank*, while a two-hop path becomes *security (RelatedTo) vault (AtLocation) bank*. These linearizations serve as knowledge statements for prompting.

We employ two prompt templates. The base template presents the question $q$ followed by five labeled choices: (A) $c_1$, (B) $c_2$, (C) $c_3$, (D) $c_4$, (E) $c_5$. The knowledge-augmented template prepends knowledge statements before the question and choices. Given a single knowledge statement $k$, the model yields a posterior distribution $\mathbf{p}^{(k)}$. For a set of $M$ statements $\mathcal{K} = \{k_1, \ldots, k_M\}$, we aggregate responses by simple averaging:

$$\bar{\mathbf{p}} = \frac{1}{M} \sum_{m=1}^{M} \mathbf{p}^{(k_m)}, \qquad \hat{y} = \arg\max_i \bar{p}_i.$$

Unless stated otherwise, all post-knowledge quantities are computed with respect to $\bar{\mathbf{p}}$.

## 3.3 ANALYSIS FRAMEWORK

Following knowledge injection, we analyze how the probability distribution evolves: specifically, how probability mass is redistributed across candidates and whether the top-ranked choice changes. Since a model's receptiveness to new information depends on its initial certainty, we stratify instances

---

[1]We interpret the model's five-way distribution as confidence allocation over the answer choices, not over the entire vocabulary.

by base confidence $\kappa^{(0)}$ into three bands: high ($\kappa^{(0)} > 0.85$), moderate ($0.65 \leq \kappa^{(0)} \leq 0.85$), and low ($\kappa^{(0)} < 0.65$). We then examine behavioral trends across these confidence strata.

This confidence-guided analysis enables us to investigate whether and to what extent LLMs incorporate KG information in augmented prompts. The stratification allows us to test whether highly confident predictions exhibit greater resistance to contradictory evidence and whether low-confidence predictions show increased susceptibility to perturbations or noise.

We evaluate three types of knowledge interventions. Supportive knowledge (or evidence) consists of statements from $\mathcal{P}_{\hat{y}^{(0)}}$ that reinforce the model's initial top choice (rank 1), regardless of its correctness. Rival knowledge comprises statements from $\mathcal{P}_r$ for ranks $r \in \{2, 3, 4, 5\}$ that challenge the base prediction by providing evidence for lower-ranked options. Finally, noisy knowledge includes random 2-hop paths sampled from the KG that are unconstrained by question or answer concepts and should ideally be disregarded by a robust model.

For every instance, knowledge condition, and confidence bin, we observe the full post-knowledge distribution $\bar{\mathbf{p}}$, per-choice probability shifts $\Delta\mathbf{p} = \bar{\mathbf{p}} - \mathbf{p}^{(0)}$, and the switch indicator $\mathbb{I}\{\hat{y} \neq \hat{y}^{(0)}\}$. Aggregating these measurements at the bin and condition level allows us to quantify three key behaviors as functions of the model's initial certainty: acceptance (increases for aligned choices and adoption of rivals when warranted), resistance (stability of the base prediction under rival evidence), and robustness to noise (limited drift of $\bar{\mathbf{p}}$ and low spurious switch rates).

## 4 EXPERIMENTAL SETUP

We conduct experiments on the COMMONSENSEQA development split, which contains 1,221 multiple-choice questions with five answer options each (Talmor et al., 2019). As our external commonsense knowledge source, we use CONCEPTNET (English, pruned) (Speer et al., 2018). Following QA-GNN Yasunaga et al. (2021a), we ground question and answer concepts to the KG and induce per-question subgraphs, computing connectivity features from direct (1-hop) and 2-hop relation paths between question and answer concepts.

We evaluate four instruction-tuned encoder–decoder LLMs without fine-tuning: **Flan-T5-small**, **Flan-T5-large**, **Llama-2-7b**, and **MISTRAL7B**. These models produce per-choice probabilities via a deterministic, decoding-free scoring procedure using teacher-forced negative log-likelihood over the full answer text, enabling direct comparison across knowledge conditions.

### 4.1 SCORING METHODOLOGY

Given a question $q$ and labeled choices $(A)\,c_1, \ldots, (E)\,c_5$, we first obtain a base belief vector $\mathbf{p}^{(0)} \in \Delta^4$ over the five choices by scoring the fixed prompt listing $q$ and all choices. The base prediction is $\hat{y}^{(0)} = \arg\max_i p_i^{(0)}$ with base confidence $\kappa^{(0)} = \max_i p_i^{(0)}$. We stratify items into three bins by $\kappa^{(0)}$: low ($< 0.65$), moderate ($0.65$–$0.85$), and high ($> 0.85$).

Within each bin, we create knowledge sets per item: supportive statements drawn from the subgraph of the current top-1 choice, rival statements drawn from the subgraphs of the rank-2 through rank-5 choices, and noisy statements sampled as random 2-hop KG paths (ten per item). For each condition, we prepend each statement $k \in \mathcal{K}$ to the same multiple-choice prompt and re-score the five answers, producing $\mathbf{p}^{(k)}$.

Scoring is performed in teacher-forcing mode without decoding. Each candidate answer $c_i$ is mapped to a fixed target sequence, tokenized and padded to length $T$ with pad token $p$. For a batch of size $B$, let $\mathbf{y} \in \mathbb{N}^{B \times T}$ be the padded targets and $\mathbf{L} \in \mathbb{R}^{B \times T \times V}$ the logits over vocabulary size $V$. The token-wise loss for item $i$ at time $t$ is $\ell_{i,t} = -\log \frac{\exp\left(L_{i,t,y_{i,t}}\right)}{\sum_{j=1}^{V} \exp(L_{i,t,j})}$. Pad positions are excluded via masking where $m_{i,t} = 1$ if $y_{i,t} \neq p$ and $m_{i,t} = 0$ otherwise. The sequence loss is $\mathcal{L}_i = \sum_{t=1}^{T} m_{i,t}\,\ell_{i,t}$, and the unnormalized candidate score is $s_i = -\mathcal{L}_i$. Scores for the five candidates are normalized with softmax to obtain the probability distribution: $p_i = \frac{\exp(s_i)}{\sum_{j=1}^{5} \exp(s_j)}$. Because teacher forcing contains no sampling or search, $p_i$ is deterministic given $(q, \mathcal{C})$ and the prepended knowledge text.

## 4.2 EVALUATION METRICS

We quantify how injected knowledge reshapes a model's belief over the five answer choices. For a question $q$ with candidates $\mathcal{C} = \{c_1, \ldots, c_5\}$, and the base distribution $\mathbf{p}^{(0)} \in \Delta^4$, when we inject a targeted set of $K$ statements for choice $c_t$, the model returns a posterior $\mathbf{p}^{(k)}$ per statement $k$, which we aggregate by averaging:

$$\bar{\mathbf{p}}^{(K)} = \frac{1}{K} \sum_{k=1}^{K} \mathbf{p}^{(k)}.$$

Let $\hat{y}^{(0)} = \arg\max_i p_i^{(0)}$ be the base top-1 choice and $\mathrm{rank}_i^{(0)}(q) \in \{1, \ldots, 5\}$ the base rank of option $c_i$. For targeted knowledge (supportive or rival), we measure ACCEPTANCE as the post-injection belief in the targeted answer candidate:

$$A(q, t) = \bar{p}_t^{(K)}.$$

We quantify probability LIFT as the change relative to the base probability:

$$\Delta(q, t) = \bar{p}_t^{(K)} - p_t^{(0)}.$$

We track ADOPTION via the indicator

$$\mathrm{Top1}(q, t) = \mathbb{I}\big[\arg\max_i \bar{p}_i^{(K)} = t\big],$$

which measures retention for supportive knowledge and conversion for rivals. For noisy (untargeted) knowledge, we measure ROBUSTNESS via the confidence the model assigns to its base top–1 ranked answer candidate before/after injecting $K$ random statements.

In addition, we report multiple–choice ACCURACY for each condition (base and post–injection), defined as the fraction of questions answered correctly:

$$\mathrm{Acc} = \frac{\#\{\, q \in \mathcal{Q} \;:\; \arg\max_i \pi_i(q) = y^\star(q)\,\}}{|\mathcal{Q}|},$$

where $\pi$ is the relevant distribution (e.g., $\mathbf{p}^{(0)}$ for the base condition or $\bar{\mathbf{p}}^{(K)}$ after targeted/noisy injection) and $y^\star(q)$ is the gold label.

## 5 RESULTS

### 5.1 TARGETED KNOWLEDGE: SUPPORT AND RIVAL EVIDENCE

We analyze how injected knowledge reshapes the model's belief distribution by grouping predictions into three confidence levels (low, moderate, and high) and evaluating five intervention settings: SupportRank1, where knowledge aligns with the model's original top prediction, and RivalRank2–5, where it supports alternatives originally ranked second to fifth. We report three metrics to capture the model's response: ACCEPTANCE, the post-injection probability assigned to the supported answer candidate; LIFT, the change in that probability relative to its baseline value, showing the direct influence of knowledge; and ADOPTION, the proportion of cases where the supported candidate becomes the new top-1 prediction, indicating a complete change in the model's decision.

Figure 1 visualizes the full ACCEPTANCE distribution (post-knowledge probability assigned to the targeted choice) for each model, stratified by base confidence (low, mid, high) and knowledge target (SUPPORT@RANK1, RIVAL@RANK$k$, $k \in \{2, 3, 4, 5\}$). Two main patterns can be observed. First, supportive evidence consistently amplifies the model's prior: for most models and strata the SUPPORT@RANK1 boxes are concentrated high on the 0–1 scale, often with upper quartiles near 1.0 for the T5 variants. Second, rival acceptance is strongly ordered by rank: RIVAL@RANK2 typically has the highest median and tightest spread among rivals, with acceptance declining progressively for ranks 3, 4, and 5. Across base-confidence strata, rivals are accepted most in the low bin and least in the high bin, indicating greater resistance when the model is initially confident.

The width of the boxes reflects this pattern. At low and moderate base confidence, the interquartile ranges (IQRs) are wider because there is more room to shift probability mass, leading to greater

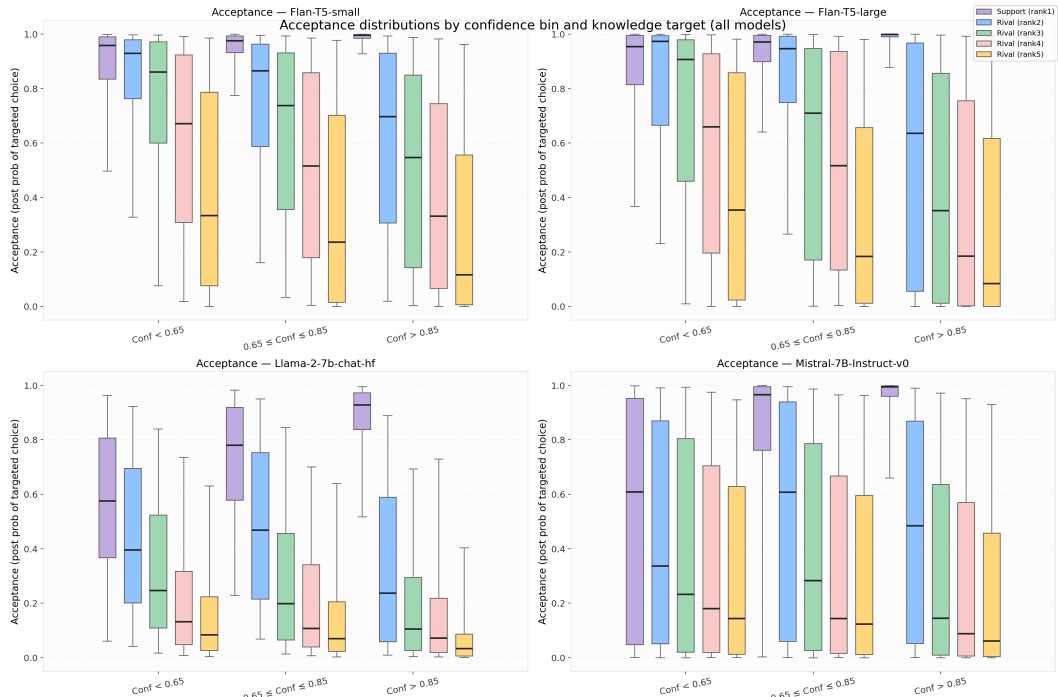

Figure 1: ACCEPTANCE distributions by base confidence and knowledge target. Each panel corresponds to a model, and boxplots show the acceptance (post-injection probability assigned to the supported answer candidate) under SUPPORT@RANK1 and RIVAL@RANK$k$ conditions. Supporting the original top prediction consistently increases confidence in that choice, while rival acceptance follows the base ranking order and decreases as base confidence grows.

variation across items. At high base confidence, rival distributions become narrower and shift downward, indicating consistent resistance to change, while SUPPORT@RANK1 distributions become more concentrated upward, showing stable reinforcement of the original choice. Across models, the T5 variants show the strongest support amplification and the clearest rank ordering. Llama-2 exhibits lower rival acceptance, while Mistral falls in between but follows the same overall trend.

Figure 2 summarizes the results using median ACCEPTANCE, highlighting two main effects: (i) a clear ranking pattern (RIVAL@2 > RIVAL@3 > RIVAL@4 > RIVAL@5), and (ii) a confidence effect, where acceptance of rival candidates decreases as base confidence increases from low to medium to high. These results confirm what the box plots suggest: LLMs strengthen the option they already prefer when given supporting evidence, and they shift toward alternative options based on how competitive those alternatives were and how uncertain the model was initially.

Across all models and confidence strata, knowledge injection produces consistent patterns in both LIFT and ADOPTION. Detailed per-model, per-bin results for LIFT and ADOPTION are provided in the Appendix tables. Supportive evidence reliably increases the model's belief in its original top choice, resulting in positive LIFT in all bins. Median gains are largest for low and moderate confidence predictions, while high-confidence predictions show smaller increases due to ceiling effects. ADOPTION rates are highest for confident base predictions and decline for moderate and low-confidence items, indicating that supportive knowledge primarily reinforces existing preferences rather than overturning decisions. Rival knowledge shows a clear rank-dependent effect on both LIFT and ADOPTION. Rank-2 rivals achieve the largest post-injection values, with values decreasing monotonically through ranks 3–5. This pattern reflects the base probability distribution over non-top options and demonstrates that models are most responsive to evidence supporting options that were already competitive. Deeper distractors (rank-4/5) rarely gain enough probability to become the top choice. Initial confidence strongly modulates both LIFT and ADOPTION for rival knowledge: low-confidence predictions are most susceptible, moderate-confidence predictions show smaller

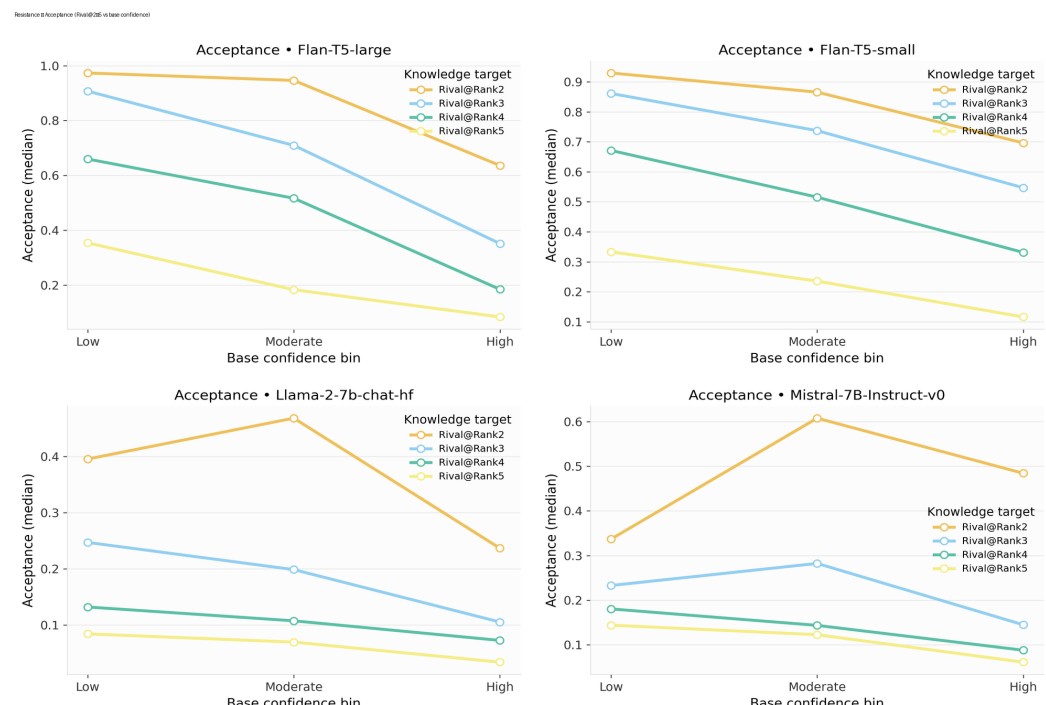

Figure 2: Median ACCEPTANCE trends. Median ACCEPTANCE is shown by base-confidence bin for each rival rank (RIVAL@2–5). The curves decrease as rival rank increases and as base confidence grows, indicating that resistance to updating depends on both factors.

effects, and high-confidence predictions are largely resistant. This aligns with the intuition that confident predictions are anchored and require stronger contradictory evidence to be overturned.

All three evaluated models exhibit the same qualitative trends: supportive knowledge amplifies belief and retention for the base answer, rival knowledge produces rank-ordered effects with rank-2 dominating, and higher base confidence increases robustness against rival interventions. Absolute magnitudes differ by model and confidence bin, but the relative patterns remain consistent.

## 5.2 NOISY KNOWLEDGE: ROBUSTNESS TO UNTARGETED INFORMATION

Figure 3 shows the global noise ROBUSTNESS, for each model and confidence level (low, medium, high), the median probability assigned to answers by their original rank (1 = top) before and after injecting $K=10$ unrelated statements. Three main patterns appear. First, the rank-1 curve drops slightly after noise, but the two lines stay nearly parallel, indicating a small reduction in confidence rather than a change in ranking. Second, lower-ranked options gain only a small amount of probability, and these gains are rarely enough to overtake the top choice. Third, the size of the effect depends on initial confidence: high-confidence items show the smallest decrease, mid-confidence items a slightly larger one, and low-confidence items the largest, although the top answer usually remains unchanged. Overall, off-topic text has little influence on the models' predictions; it behaves more like a weak regularizer on probability distribution than a cause of rank changes.

To examine whether noise can weaken the effect of useful evidence, we start with correct supporting knowledge (1–2 hop chains linking the question concepts to the correct answer candidate) and gradually add unrelated statements at $\frac{1}{4}\times$, $\frac{1}{2}\times$, $1\times$, $2\times$, $3\times$, and $5\times$ the number of support statements. Figure 4 shows ACCURACY, separating answer candidates based on whether each model's initial prediction was correct. For candidates that were already correct, using only support keeps accuracy near its maximum, while adding noise causes a smooth and steady decline, indicating signal dilution rather than active misdirection. For initially incorrect candidates, support alone produces the largest improvements (many predictions flip to the correct label). As noise increases, accuracy decreases

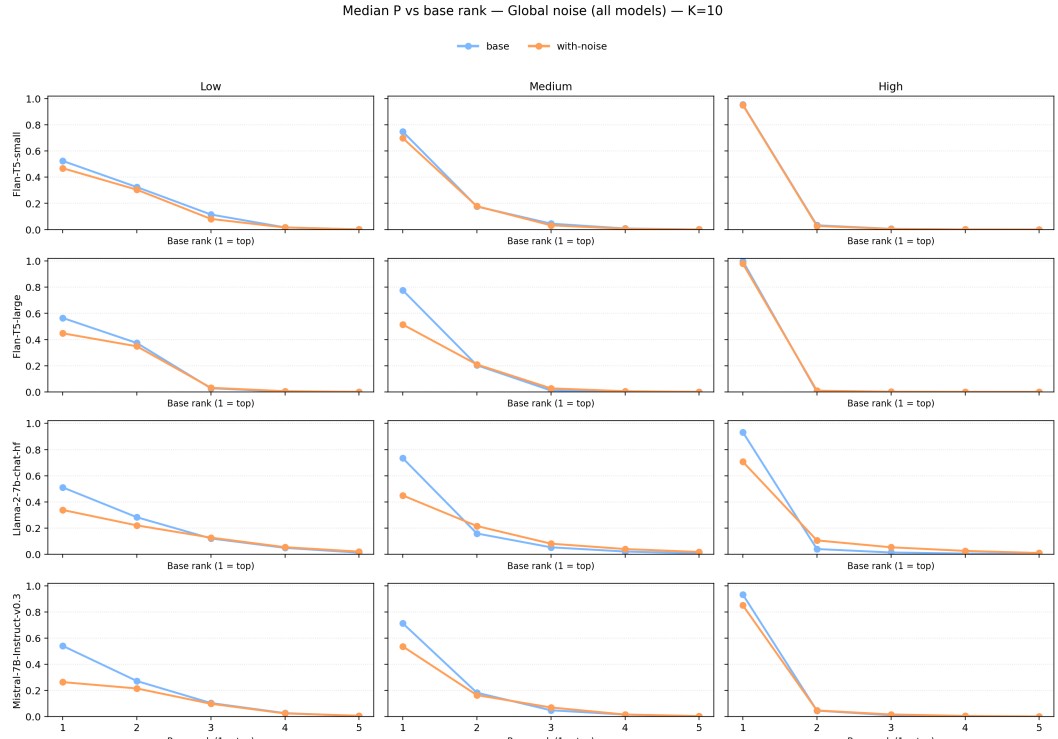

Figure 3: Global noise ROBUSTNESS (K=10). Median probability assigned to each answer as a function of its base rank (1 = top) for all models and base–confidence strata (low/mid/high). Blue = base (no KG), orange = with 10 untargeted (global) statements. Noise slightly lowers the top–1 confidence while largely preserving the rank ordering; attenuation is smallest at high base confidence.

steadily toward the no-KG baseline: when irrelevant text dominates the prompt, the model tends to ignore it, but its weight reduces the influence of helpful evidence, limiting corrections. Across models (Flan-T5-small/large, Llama-2-7b, Mistral-7B), these trends are stable: untargeted noise rarely induces top-1 changes by itself, yet in aggregate it dilutes the benefits of targeted support as its volume eclipses the signal. Complete tables summarizing median base confidence under different noise levels ($K=10, 20, 30$) as well as full accuracy breakdowns for all models and confidence strata are provided in the Appendix (Tables 2–3).

Overall, these analyses reveal clear behavioral regularities: LLMs amplify their priors when given supportive evidence, update most readily toward rival options that were already competitive (rank–2), and resist or disregard rival evidence proportionally to their initial confidence. These trends hold consistently across model families and confidence strata, offering practical guidance for knowledge curation in KG-augmented question answering. With untargeted global noise, models typically retain their base choice and exhibit only modest median drops in confidence. However, when correct support is paired with increasing amounts of added noise (from $1/4\times$ up to $5\times$ the support size), accuracy declines monotonically: the injected noise progressively washes out the benefit of the support, especially on instances the model initially mispredicted.

## 6 CONCLUSION

We analyze the effect of knowledge based on two factors: the model's base confidence and the rank of the supported answer candidate. Three types of knowledge are considered. Supportive evidence consists of KG statements that reinforce the model's original top choice (rank 1), regardless of correctness. Rival knowledge provides evidence for lower-ranked options, challenging the base prediction, while noisy knowledge consists of random 2-hop paths from the KG that are unrelated to the question or answer candidates and should ideally be ignored by the model. Supportive

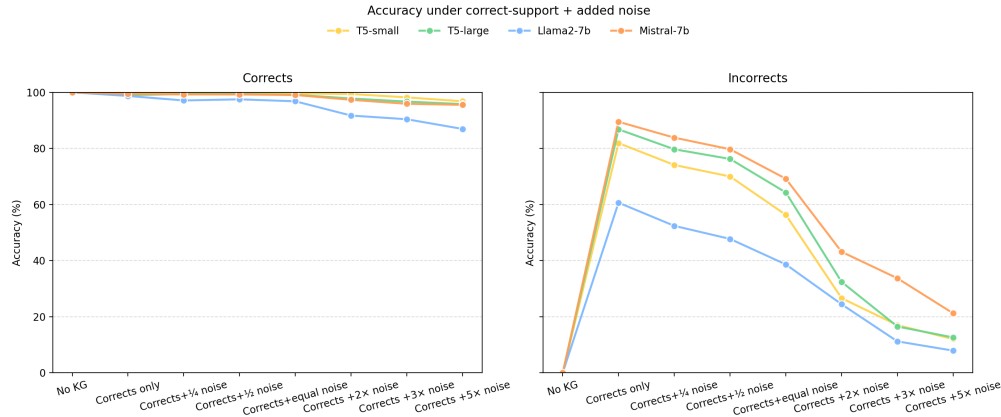

Figure 4: ACCURACY with correct support plus increasing untargeted noise. Items are split by each model's initial correctness. Starting from correct supporting knowledge (1–2 hop QC→AC paths), we add global noise at $\frac{1}{4}\times, \frac{1}{2}\times, 1\times, 2\times, 3\times, 5\times$ the support count. Accuracy declines smoothly as noise grows: untargeted text is mostly ignored in isolation, but in aggregate it dilutes the influence of the supportive evidence, reducing conversions on initially incorrect items and slightly lowering accuracy on initially correct items.

evidence almost always increases the probability of the top-1 choice, with minimal additional gain when confidence is already high, showing a ceiling effect. Rival knowledge is most effective for near alternatives: rank-2 and rank-3 candidates gain the most probability and have the highest switch rates at low to moderate confidence, whereas rank-4 and rank-5 candidates show smaller, inconsistent gains and are rarely adopted. Acceptance decreases with lower internal preference (rank1 > rank2 > rank3 > rank4 > rank5) and increases as base confidence decreases. Untargeted noise has a small effect, slightly lowering the top probability for uncertain candidates. When mixed with supportive evidence, the impact depends on the signal-to-noise ratio: correct top choices remain stable, while initially incorrect candidates are more affected.

These results support a confidence-aware intervention approach. For high-confidence predictions, provide short confirmatory evidence or do not intervene, as re-ranking is unlikely to succeed. For moderate or low-confidence predictions, focus on near alternatives (rank-2 or rank-3) with concise supportive statements and avoid distant options (rank-4 or rank-5). Noise should be limited and relevant; when combined with support, maintain a high signal-to-noise ratio to preserve the effect on initially incorrect candidates.

This paper focuses on confidence response rather than overall accuracy to understand how knowledge changes predictions and which interventions are effective. The findings suggest that providing concise, relevant evidence to near alternatives, only when the model's confidence is low or moderate, is the most reliable strategy for improving KG-augmented question answering.

## 7 FUTURE WORK

Future work includes several directions. First, we will explore rank-aware confidence control that penalizes probability gains based on a rival's base-rank distance (e.g., per-rank temperatures, prior gates, or calibration losses). Second, we plan a systematic noise taxonomy and ablation study i.e., covering off-topic facts, near-miss rivals, and paths that start from random neighbors but reach answer candidates, to quantify their differential effects. Third, we will tune prompts and decoding (instruction variants, temperature/nucleus settings, self-consistency) to improve robustness and calibration. In parallel, we will refine evidence selection and weighting to adapt $K$ per instance, detect contradictions, and balance support versus rival signals. We also aim to incorporate richer calibration and diagnostics (ECE, Brier score, counterfactual acceptance) and to extend coverage across additional model families and datasets to evaluate transfer under domain shift.

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

## A  APPENDIX

This appendix compiles the full quantitative tables that underpin our analyses. Table 1 reports targeted–knowledge outcomes (Support@Rank1 and Rival@Rank2–5) for each model and confidence stratum, including Acceptance, Lift, Adoption, and the Base Confidence (Median [Q1–Q3]). Table 2 summarizes robustness to untargeted noise by showing the base top–1 confidence and its post-injection value under $K \in \{10, 20, 30\}$ random statements. Table 3 presents multiple-choice accuracy when correct-support knowledge is mixed with increasing amounts of random noise.

**Reporting conventions.**  Unless noted otherwise, Acceptance, Lift, and Base Confidence are reported as Median [Q1–Q3], while Adoption is Mean $\pm$ SD. Confidence strata are defined by the base top–1 probability $\kappa^{(0)}$: *low* $< 0.65$, *mid* $0.65$–$0.85$, *high* $> 0.85$. For each (model, bin), $N$ is the count of items with usable extractions under all required conditions.

Table 1 summarizes targeted-knowledge effects across models and confidence strata. Support@Rank1 consistently amplifies the model's prior—its post-injection Acceptance exceeds the Base Confidence median and Lift is positive in every bin— with the biggest gains in low/mid confidence and smaller gains at high confidence (ceiling effects). Rival evidence follows a strict rank ordering: Rival@Rank2 yields the highest Acceptance and Adoption among rivals, declining monotonically through ranks 3–5, mirroring the base probability mass on those distractors. Susceptibility is confidence-dependent: rival Lift/Adoption is largest in the low bin, smaller in the mid bin, and smallest in the high bin, indicating that strongly confident base predictions are hardest to overturn.

Table 2 shows that injecting random facts reduces confidence in the base top–1 choice, but the drop is generally modest and tightly governed by initial certainty. In the high bin, Flan-T5 models are notably robust (e.g., small median declines from $95-100\%$ to $\approx 98\%$), Mistral degrades more, and Llama shows the largest high-bin dip. In the low and mid bins, medians fall more substantially across all models, yet remain well above chance (20%), indicating that models mostly discount off-topic content rather than collapsing their belief. Increasing noise volume from 10 to 20 to 30 statements yields little additional degradation in the medians (often accompanied by wider IQRs), suggesting a saturation effect where extra random facts are further down-weighted.

Table 3 separates items the model initially gets correct vs. incorrect (under "No KG") and tracks accuracy as we add the correct-support knowledge and then mix in increasing amounts of random noise. On the **Corrects** split, adding the correct-support alone leaves accuracy essentially unchanged (all models $\approx$98–100%), and even substantial noise produces only modest erosion: by $5\times$ noise, Flan-T5-small drops from $99.8\% \rightarrow 96.8\%$, Flan-T5-large $99.0\% \rightarrow 95.8\%$, Mistral $99.4\% \rightarrow 95.5\%$, while Llama2 shows a non-monotonic path but remains high overall ($98.7\% \rightarrow 96.9\%$). This indicates that when the base prediction is already correct, supportive evidence is robust to untargeted facts until the noise-to-signal ratio becomes extreme.

On the **Incorrects** split, providing only the correct-support substantially rescues many cases (e.g., Flan-T5-large: $0\% \rightarrow 86.8\%$; Mistral: $0\% \rightarrow 89.5\%$), confirming that targeted, on-graph evidence can overturn initial mistakes. However, as random noise grows, these gains collapse sharply: at $5\times$ noise, accuracy falls to 12.0% (Flan-T5-small), 12.6% (Flan-T5-large), 7.9% (Llama2), and 21.2% (Mistral). In short, when the base decision is wrong, support can correct it, but heavy off-topic content progressively washes out that signal; when the base decision is right, accuracy remains high and degrades only gradually under the same noise levels.

Table 1: Knowledge intervention effects. Base Confidence is Median [Q1–Q3]. Acceptance/Lift are Median [Q1–Q3]; Adoption is Mean ± SD.

| Model | Conf. Bin | N | Base % (Med [Q1–Q3]) | Scenario | Acceptance % (Med [Q1–Q3]) | Lift % (Med [Q1–Q3]) | Adoption % (Mean ± SD) |
|---|---|---|---|---|---|---|---|
| T5-SMALL | low | 423 | 52.27[45.65–59.08] | Support@Rank1 | 95.86 [83.48–99.08] | 38.68 [29.03–47.78] | 99.29 ± 8.4 |
| | | | | Rival@Rank2 | 92.93 [76.37–97.96] | 57.77 [43.41–65.77] | 89.6 ± 30.56 |
| | | | | Rival@Rank3 | 86.1 [60.07–97.21] | 71.67 [47.98–83.45] | 83.69 ± 36.99 |
| | | | | Rival@Rank4 | 67.12 [30.84–92.36] | 62.76 [27.11–87.62] | 70.21 ± 45.79 |
| | | | | Rival@Rank5 | 33.35 [7.64–78.69] | 32.29 [7.38–76.82] | 47.75 ± 50.01 |
| T5-SMALL | mid | 342 | 75.42 [70.1–80.84]% | Support@Rank1 | 97.56 [93.3–99.4] | 20.5 [14.87–26.29] | 99.12 ± 9.34 |
| | | | | Rival@Rank2 | 86.55 [58.76–96.36] | 66.04 [40.06–77.33] | 82.46 ± 38.09 |
| | | | | Rival@Rank3 | 73.72 [35.64–93.14] | 67.88 [29.58–86.92] | 72.51 ± 44.71 |
| | | | | Rival@Rank4 | 51.56 [17.99–85.9] | 50.34 [15.41–82.74] | 57.02 ± 49.58 |
| | | | | Rival@Rank5 | 23.58 [1.53–70.26] | 23.56 [1.24–70.15] | 40.35 ± 49.13 |
| T5-SMALL | high | 456 | 95.47 [90.66–98.47] | Support@Rank1 | 99.56 [98.54–99.91] | 3.1 [0.47–7.17] | 100.0 ± 0.0 |
| | | | | Rival@Rank2 | 69.64 [30.68–93.07] | 65.39 [26.55–87.75] | 64.47 ± 47.91 |
| | | | | Rival@Rank3 | 54.69 [14.3–85.02] | 53.24 [13.74–83.49] | 55.04 ± 49.8 |
| | | | | Rival@Rank4 | 33.14 [6.69–74.53] | 32.49 [6.36–73.72] | 42.11 ± 49.43 |
| | | | | Rival@Rank5 | 11.63 [0.69–55.6] | 11.63 [0.66–55.58] | 29.82 ± 45.8 |
| T5-LARGE | low | 97 | 55.92 [49.76–59.71] | Support@Rank1 | 95.36 [81.5–99.58] | 37.36 [22.02–43.81] | 94.85 ± 22.23 |
| | | | | Rival@Rank2 | 97.34 [66.55–99.46] | 54.18 [31.69–62.08] | 86.6 ± 34.24 |
| | | | | Rival@Rank3 | 90.71 [46.01–97.91] | 77.97 [37.64–91.7] | 75.26 ± 43.38 |
| | | | | Rival@Rank4 | 65.98 [19.65–92.82] | 61.35 [19.35–91.38] | 61.86 ± 48.83 |
| | | | | Rival@Rank5 | 35.42 [2.38–85.81] | 35.42 [2.37–85.09] | 45.36 ± 50.04 |
| T5-LARGE | mid | 97 | 76.94 [71.7–81.09] | Support@Rank1 | 97.14 [89.92–99.56] | 17.74 [13.48–23.01] | 96.91 ± 17.4 |
| | | | | Rival@Rank2 | 94.62 [74.89–99.27] | 70.82 [55.06–79.36] | 84.54 ± 36.34 |
| | | | | Rival@Rank3 | 70.94 [17.05–94.77] | 70.39 [13.79–90.63] | 62.89 ± 48.56 |
| | | | | Rival@Rank4 | 51.68 [13.45–93.64] | 51.58 [11.07–93.64] | 55.67 ± 49.94 |
| | | | | Rival@Rank5 | 18.35 [1.2–65.75] | 18.33 [1.2–65.64] | 38.14 ± 48.83 |
| T5-LARGE | high | 1027 | 99.89 [98.52–99.99] | Support@Rank1 | 99.88 [99.0–99.98] | 0.0 [-0.15–0.22] | 99.03 ± 9.82 |
| | | | | Rival@Rank2 | 63.62 [5.58–96.81] | 61.42 [4.96–92.82] | 55.6 ± 49.71 |
| | | | | Rival@Rank3 | 35.16 [1.24–85.62] | 35.05 [0.96–85.0] | 44.3 ± 49.7 |
| | | | | Rival@Rank4 | 18.5 [0.25–75.62] | 18.5 [0.23–75.62]% | 36.22 ± 48.09 |
| | | | | Rival@Rank5 | 8.42 [0.06–61.78] | 8.39 [0.05–61.78]% | 31.06 ± 46.3 |
| LLAMA2-7B | low | 617 | 50.66 [42.33–57.56] | Support@Rank1 | 57.59 [36.72–80.63] | 3.07 [-10.35–26.37] | 74.88 ± 43.41 |
| | | | | Rival@Rank2 | 39.55 [20.14–69.55] | 10.42 [-3.83–40.41] | 49.92 ± 50.04 |
| | | | | Rival@Rank3 | 24.69 [10.94–52.36] | 12.01 [0.0–40.24] | 35.66 ± 47.94 |
| | | | | Rival@Rank4 | 13.18 [4.79–31.73] | 7.53 [0.0–25.23] | 22.53 ± 41.81 |
| | | | | Rival@Rank5 | 8.39 [2.66–22.42] | 6.12 [0.86–20.79] | 15.88 ± 36.58 |
| LLAMA2-7B | mid | 333 | 73.92 [69.36–79.52] | Support@Rank1 | 78.0 [57.9–91.9] | 1.31 [-14.65–15.78] | 87.09 ± 33.58 |
| | | | | Rival@Rank2 | 46.84 [21.52–75.37] | 31.99 [1.32–55.06] | 55.86 ± 49.73 |
| | | | | Rival@Rank3 | 19.85 [6.55–45.62] | 14.64 [0.0–40.77] | 29.73 ± 45.78 |
| | | | | Rival@Rank4 | 10.72 [3.96–34.13] | 8.47 [0.77–30.58] | 21.62 ± 41.23 |
| | | | | Rival@Rank5 | 6.92 [2.41–20.62] | 6.04 [1.5–19.87] | 15.92 ± 36.64 |
| LLAMA2-7B | high | 271 | 92.94 [89.68–95.85] | Support@Rank1 | 92.84 [83.83–97.29] | -0.24 [-7.8–2.99] | 97.05 ± 16.96 |
| | | | | Rival@Rank2 | 23.66 [5.92–58.95] | 18.76 [1.19–53.76] | 35.06 ± 47.8 |
| | | | | Rival@Rank3 | 10.51 [2.74–29.48] | 8.66 [0.61–27.25] | 16.61 ± 37.28 |
| | | | | Rival@Rank4 | 7.23 [1.98–21.9] | 6.5 [1.2–21.8] | 13.65 ± 34.4 |
| | | | | Rival@Rank5 | 3.35 [0.67–8.64] | 3.07 [0.46–8.61] | 6.27 ± 24.29 |
| MISTRAL7B | low | 390 | 52.23 [45.56–58.11] | Support@Rank1 | 60.9 [4.88–95.24] | 0.0 [-39.56–32.26] | 62.56 ± 48.46 |
| | | | | Rival@Rank2 | 33.69 [5.14–86.96] | 0.0 [-18.38–54.85] | 44.36 ± 49.74 |
| | | | | Rival@Rank3 | 23.3 [2.06–80.52] | 11.3 [-5.68–70.2] | 42.82 ± 49.55 |
| | | | | Rival@Rank4 | 18.05 [1.94–70.55] | 15.79 [0.0–66.66] | 39.49 ± 48.95 |
| | | | | Rival@Rank5 | 14.41 [1.26–62.88] | 12.84 [0.0–61.79] | 35.64 ± 47.96 |
| MISTRAL7B | mid | 320 | 75.91 [70.5–80.18] | Support@Rank1 | 96.51 [76.16–99.5] | 15.87 [0.0–29.23] | 85.94 ± 34.82 |
| | | | | Rival@Rank2 | 60.77 [5.99–93.92] | 41.78 [-3.31–70.33] | 55.0 ± 49.83 |
| | | | | Rival@Rank3 | 28.24 [2.63–78.65] | 23.97 [-0.42–73.85] | 41.25 ± 49.31 |
| | | | | Rival@Rank4 | 14.36 [1.61–66.75] | 12.55 [0.0–59.7] | 35.31 ± 47.87 |
| | | | | Rival@Rank5 | 12.28 [1.14–59.57] | 11.35 [0.11–57.71] | 31.87 ± 46.67 |
| MISTRAL7B | high | 511 | 95.43 [91.0–98.58] | Support@Rank1 | 99.46 [96.0–99.89] | 2.9 [0.03–10.92] | 96.67 ± 17.95 |
| | | | | Rival@Rank2 | 48.44 [5.3–86.91] | 41.43 [0.0–76.36] | 50.49 ± 50.05 |
| | | | | Rival@Rank3 | 14.5 [0.96–63.7] | 12.98 [0.0–59.98] | 34.64 ± 47.63 |
| | | | | Rival@Rank4 | 8.8 [0.64–57.0] | 8.52 [0.06–55.24] | 31.12 ± 46.34 |
| | | | | Rival@Rank5 | 6.13 [0.4–45.79] | 5.81 [0.18–45.17] | 26.81 ± 44.34 |

Table 2: Impact of random noisy knowledge on the model's base-prediction confidence. Values are Median [Q1–Q3].

| Model | Confidence Bin | N | Base Conf. (Median [Q1–Q3]) | Post@10 Noise | Post@20 Noise | Post@30 Noise |
|---|---|---|---|---|---|---|
| Flan-T5-small | low | 422 | 52.43 [45.37–58.59]% | 46.82 [29.13–63.16]% | 46.23 [28.32–63.25]% | 47.07 [28.07–62.69]% |
| | mid | 344 | 74.78 [69.85–80.44]% | 69.85 [52.61–83.07]% | 69.81 [53.02–82.96]% | 68.85 [53.38–82.63]% |
| | high | 455 | 95.45 [90.67–98.52]% | 95.11 [88.19–98.79]% | 94.7 [87.46–98.82]% | 94.56 [87.66–98.82]% |
| Flan-T5-large | low | 99 | 56.5 [50.28–60.25]% | 44.77 [13.92–72.34]% | 43.51 [16.72–72.46]% | 43.66 [14.73–72.48]% |
| | mid | 98 | 77.63 [72.43–80.68]% | 51.46 [29.66–81.92]% | 59.1 [26.55–82.34]% | 56.39 [26.12–82.31]% |
| | high | 1024 | 99.89 [98.5–99.99]% | 97.97 [84.93–99.79]% | 97.98 [84.27–99.78]% | 97.77 [83.99–99.77]% |
| Llama-2-7b-chat-hf | low | 617 | 51.09 [43.06–59.48]% | 33.87 [17.21–55.74]% | 34.42 [17.53–56.28]% | 34.4 [17.16–56.46]% |
| | mid | 333 | 73.43 [67.25–80.51]% | 44.94 [25.08–67.91]% | 44.21 [24.07–70.35]% | 46.12 [24.84–69.46]% |
| | high | 271 | 93.28 [89.14–96.13]% | 70.69 [49.85–86.86]% | 70.64 [49.03–86.64]% | 71.14 [50.9–86.43]% |
| Mistral-7B-Instruct-v0.3 | low | 390 | 54.09 [45.88–64.28]% | 26.39 [8.74–58.34]% | 27.41 [10.2–57.8]% | 27.97 [10.61–55.55]% |
| | mid | 320 | 71.32 [60.28–81.96]% | 53.65 [23.47–75.9]% | 52.09 [24.05–75.14]% | 50.32 [22.98–75.08]% |
| | high | 511 | 93.27 [83.82–97.82]% | 85.04 [58.54–96.44]% | 85.35 [56.48–96.08]% | 84.24 [56.84–96.16]% |

Table 3: Accuracy with correct-support knowledge mixed with increasing random noise.

| Model | Question type | No KG | Corrects only | + $\frac{1}{4}$ noise | + half noise | + same noise | + double noise | + triple noise | + 5× noise |
|---|---|---|---|---|---|---|---|---|---|
| T5-Flan-small | Corrects | 100.0% | 99.80% | 99.60% | 99.60% | 99.60% | 99.40% | 98.20% | 96.80% |
| T5-Flan-large | Corrects | 100.0% | 99.00% | 99.38% | 99.38% | 99.10% | 97.80% | 96.69% | 95.80% |
| Llama2-7b | Corrects | 100.0% | 98.70% | 97.10% | 97.50% | 96.80% | 91.72% | 90.4% | 96.94% |
| Mistral-7b | Corrects | 100.0% | 99.43% | 99.24% | 99.23% | 99.05% | 97.32% | 95.90% | 95.52% |
| T5-Flan-small | Incorrects | 0.0% | 81.90% | 74.10% | 70.00% | 56.32% | 26.60% | 17.00% | 12.00% |
| T5-Flan-large | Incorrects | 0.0% | 86.80% | 79.70% | 76.24% | 64.30% | 32.40% | 16.50% | 12.60% |
| Llama2-7b | Incorrects | 0.0% | 60.65% | 52.39% | 47.73% | 38.64% | 24.43% | 11.20% | 7.90% |
| Mistral-7b | Incorrects | 0.0% | 89.50% | 83.84% | 79.7% | 69.2% | 43.13% | 33.7% | 21.2% |

