# OpenReview forum: "When Do LLMs Listen? Confidence-Guided Knowledge Acceptance in LLMs"
_ICLR.cc/2026/Conference — ICLR 2026 Conference Withdrawn Submission_

### Official Review · Reviewer_yVdB · 2025-10-20

**Soundness:** 2
**Presentation:** 2
**Contribution:** 1
**Rating:** 2
**Confidence:** 4

**Summary:**

This paper studies whether LLMs “listen” to the intended target when multiple cues are mixed in the prompt. Using knowledge graph (KG)–derived supportive, rival, and noisy evidence, the authors analyze how conflicting cues affect model obedience and claim that instruction-following can be fragile under such multi-cue settings.

**Strengths:**

- The paper isolates cue-level behavior instead of only measuring QA accuracy, providing a clear behavioral perspective.

- The experimental setup is controlled and easy to reproduce.

- The motivation is reasonable, as cue robustness is related to instruction-following reliability.

**Weaknesses:**

- Very similar behavior has already been studied in noisy RAG and KG-based robustness work, where relevant and noisy evidence compete; the core phenomenon is not new.

- KG literature already includes studies on misleading knowledge, noise injection, and evidence competition, so the contribution over prior KG/RAG robustness papers is unclear.

- The paper provides observation only, without deeper analysis or actionable solution, limiting its impact.

**Questions:**

See above.

---

### Official Review · Reviewer_xML7 · 2025-10-20

**Soundness:** 2
**Presentation:** 2
**Contribution:** 1
**Rating:** 2
**Confidence:** 4

**Summary:**

This paper studies when LLMs “listen” to external knowledge injected from a KG during multiple-choice QA. The authors bin questions by the model’s baseline confidence and then add (i) supportive paths for the model’s current top choice, (ii) rival paths aligned to other choices, or (iii) random noisy paths. They average posteriors across statements and report acceptance/lift/adoption patterns across bins. The main claim is that high-confidence predictions resist change, mid/low-confidence predictions shift more, and noise slightly dilutes probabilities.

**Strengths:**

1. Clear experimental framing: supportive vs. rival vs. noisy knowledge is an intuitive setup.
2. Confidence-stratified analysis is easy to understand and can be a useful lens.

**Weaknesses:**

1. The mapping from question/answers to KG concepts is not specified with enough detail to be reproducible.
2. The paper seems to presume that incorrect options are linked to the question in the KG, but does not justify why that should hold or how often it fails.
3. There is no human or automatic check that extracted paths can actually support the model's reasoning of the supportive statements.
4. Evaluation breadth is narrow: one dataset (CommonsenseQA) and a small model set; no tests on other QA tasks, other KGs.

**Questions:**

1. How exactly do you map question spans and each answer option to KG nodes?
2. For each bin and choice rank, what percentage of (question, option) pairs have at least one extracted path?
3. How do you verify that a “supportive” path truly supports the correct answer rather than introducing bias/noise?

---

### Official Review · Reviewer_Gxnj · 2025-10-30

**Soundness:** 3
**Presentation:** 3
**Contribution:** 2
**Rating:** 4
**Confidence:** 3

**Summary:**

This paper studies when LLMs accept or resist external knowledge injected from a Knowledge Graph (KG) during in-context learning for multiple-choice QA. The authors propose a confidence-guided analysis: partition baseline model predictions into high/medium/low-confidence bands, then inject (i) supportive knowledge that favors the baseline top-1 option, (ii) rival knowledge aligned with alternative options (rank-2/3/4/5), and (iii) noisy off-topic statements. Using teacher-forcing probabilities, they quantify effects via Acceptance, Lift, and Adoption (switch) rates and assess robustness under noise.

**Strengths:**

* Confidence-stratified acceptance is a clean framing that reveals stable patterns (support reinforces; near-rank rivals are most potent; high-confidence states are robust).
* Well-controlled interventions. Separation of support/rival/noise and rank-aware rivaling is thoughtful. The acceptance/boost/adoption model is intuitive and reproducible.
* Trends hold across several popular LLMs, which increases reliability.

**Weaknesses:**

* Experiments focus on commonsense MCQA with ConceptNet-style paths. It is unclear if findings transfer to open-ended QA, multi-hop reasoning, or to other KGs (e.g., Wikidata) and domains.
* Acceptance may depend on how KG paths are verbalized. The paper does not provide sensitivity analyses to paraphrasing, prompt length, prompt ordering, or number of injected paths.
* The paper diagnoses when to inject knowledge but does not propose or validate a confidence-gated selection/prompting policy that demonstrably improves QA. Missing a natural next step that would elevate impact.
* “Off-topic” noise is a clean control but less realistic than semantically plausible yet subtly misleading evidence. Adversarial or conflicting knowledge is under-explored.

**Questions:**

Please refer to the Weaknesses.

---

### Official Review · Reviewer_p6Zj · 2025-11-06

**Soundness:** 3
**Presentation:** 3
**Contribution:** 2
**Rating:** 2
**Confidence:** 4

**Summary:**

This paper investigates how large language models (LLMs) respond to injected external knowledge—particularly from knowledge graphs (KGs)—in multiple-choice question answering tasks. Rather than focusing on accuracy alone, the authors introduce a confidence-guided framework that stratifies model predictions into high, moderate, and low confidence levels, and analyzes how different forms of knowledge (supportive, rival, and noisy) influence model predictions. Experiments on CommonsenseQA with multiple instruction-tuned LLMs reveal consistent behavioral patterns: high-confidence predictions tend to resist change, while low- and mid-confidence predictions are more susceptible to knowledge interventions. The findings offer practical insights into the design of KG-augmented prompting strategies.

**Strengths:**

1. The paper uses a confidence-aware analysis to provide a more detailed understanding of when and how LLMs integrate external knowledge.

2. The experiments are thorough and well-controlled, covering multiple model families and knowledge types, and the results reveal consistent empirical patterns.

**Weaknesses:**

1. The main method and findings are largely expected—e.g. high-confidence predictions tend to resist change while low- and mid-confidence predictions are more susceptible—offering limited new insight.

2. While a non-trivial number of prior works have explored when LLMs accept or disregard external knowledge, the paper does not clearly situate itself within this line of research.

**Questions:**

Please refer to the above section.

---

### Note · Authors · 2025-12-22

I have read and agree with the venue's withdrawal policy on behalf of myself and my co-authors.